# Complications and Risk Factors in Patients with Soft Tissue Sarcoma of the Extremities Treated with Radiotherapy

**DOI:** 10.3390/cancers16111977

**Published:** 2024-05-23

**Authors:** Arthur Lebas, Clara Le Fevre, Waisse Waissi, Isabelle Chambrelant, David Brinkert, Georges Noel

**Affiliations:** 1Radiotherapy Department, Institut de Cancérologie Strasbourg Europe (ICANS), 17 Rue Albert Calmette, BP 23025, 67033 Strasbourg, France; a.lebas@icans.eu (A.L.); c.lefevre@icans.eu (C.L.F.); i.chambrelant@icans.eu (I.C.); 2Radiotherapy Department, Léon Bérard Center, 28 Rue Laennec, 69008 Lyon, France; waisse.waissi@lyon.unicancer.fr; 3Orthopedic Surgery Department, University Hospital of Hautepierre, 1 Rue Molière, 67200 Strasbourg, France; david.brinkert@chru-strasbourg.fr; 4Faculty of Medicine, Strasbourg University, 4 Rue Kirschleger, 67000 Strasbourg, France; 5Radiobiology Laboratory, Centre Paul Strauss, IIMIS—Imagerie Multimodale Integrative en Santé, ICube, Strasbourg University, 67000 Strasbourg, France

**Keywords:** soft tissue sarcoma, extremity, radiotherapy, limb-sparing surgery, prognostic factors, complications

## Abstract

**Simple Summary:**

This study assessed the overall complications in 169 patients treated for extremity soft tissue sarcomas (ESTS) with a multimodal treatment involving radiotherapy and surgery. Risk factors for complications were identified, including postoperative, acute, and chronic radiotherapy-related complications, aiming to optimize treatment strategies to reduce morbidity. Multimodal treatment of ESTSs demonstrated excellent tolerance, with manageable side effects.

**Abstract:**

Introduction: Soft tissue sarcomas of the extremities (ESTSs) pose significant challenges in treatment and management due to their diverse nature and potential complications. This study aimed to assess complications associated with multimodal treatments involving surgery and radiotherapy (RT) and to identify potential risk factors. Methods: We retrospectively analyzed nonmetastatic ESTS patients treated with surgery and pre- or post-operative RT between 2007 and 2020 in Strasbourg, France. Complications, including wound complications (WCs), lymphedema, acute and chronic RT-related complications, and fractures, were meticulously evaluated. Results: A total of 169 patients diagnosed with localized ESTSs were included, with a median age of 64 years (range 21–94 years). ESTSs primarily occurred proximally (74.6%) and in the lower limbs (71%). The median follow-up was 5.5 years. WCs occurred in 22.5% of patients, with proximal and lower extremity tumors being significant risk factors. Acute RT-related complications included radiodermatitis, with grade ≥ 2 occurring in 43.1% of patients, which was associated with superficial tumors. Three patients had an edema grade ≥ 2. Chronic complications included telangiectasias (21.7%) and fibrosis (38.7%), with higher rates associated with larger PTVs and higher RT doses, respectively. Fractures occurred in 5 patients, mainly in the tibia (40%). Conclusions: Multimodal treatment of ESTSs demonstrated excellent tolerance, with manageable side effects. Numerous risk factors have been highlighted, providing insights for optimizing treatment strategies and enhancing patient care in this rare disease.

## 1. Introduction

Soft tissue sarcomas (STSs), which account for approximately 1% of all adult solid malignancies, encompass a diverse array of tumors with more than 80 histologic subtypes originating in mesenchymal tissues [1]. They are predominantly located in the extremities, with the lower limbs being the most common site (60%) [1,2,3]. The standard treatment for locally advanced extremity soft tissue sarcoma (ESTS) involves en bloc surgery with at least negative margins. Radiation therapy (RT) is recommended for patients with ESTSs exhibiting intermediate- and high-risk features and fulfilling criteria related to size, depth, and unhealthy margins [4,5].

Prior studies have addressed complications related to surgery based on the timing of RT, where wound complications (WCs) were found to occur at least twice as often with preoperative RT [6]. In contrast, long-term complications such as fibrosis and edema were more frequent in patients treated with postoperative RT, potentially affecting functional outcomes [7]. Only a few studies have concurrently analyzed acute and long-term radiation-induced side effects. Among them, research exclusively dedicated to ESTSs is notably scarce, with even fewer prospective randomized studies available [8,9]. Moreover, the grading of side effects using the Common Terminology Criteria for Adverse Events (CTCAE) or the Radiation Therapy Oncology Group (RTOG) toxicity scale has often been underutilized [10,11]. This dearth of data has posed challenges in comparing series, particularly those employing modern radiotherapy techniques such as intensity-Modulated Radiation Therapy (IMRT) or hypofractionated schedules. Among these complications, a variety of different risk factors have been identified across studies [12,13,14].

The primary objective of the present study was to assess associations between toxicities and multimodal treatment involving a combination of surgery and RT in patients diagnosed with nonmetastatic ESTSs at our institution. Additionally, we aimed to compare our results with those documented in the literature.

## 2. Materials and Methods

### 2.1. Study Population

We conducted a retrospective monocentric study on adult patients with extremity soft tissue sarcomas treated surgically with radiotherapy, including 169 patients from 2007 to 2020.

The inclusion and exclusion criteria have already been reported in a previous article [15]. Regarding the indications for RT based on tumor characteristics, RT for grade 1 sarcoma was prescribed if one of the following conditions was met: tumor size exceeding 5 cm, R1 surgical margins, or non-monobloc surgery.

For R0 margins, RT was prescribed after a multidisciplinary discussion considering the following factors: tumor size exceeding 5 cm, grade 2–3, and deep tumor. Curative RT was prescribed for patients with R2 margins following maximal resection by the surgeon when further excision was not feasible.

### 2.2. Treatment and Irradiation Technique

Treatment techniques comprised 2D-RT, 3D-RT, or IMRT utilizing helical tomotherapy. Both preoperative and postoperative RT were utilized. Normofractionated, hypofractionated, or hyperfractionated regimens were utilized. To facilitate comparisons among the various schedules due to the different fractionation schemes, doses were converted to BEDGy_4_ or BEDGy_10_ [16]. The most used RT regimens were 50 Gy in 25 fractions of 2 Gy for patients with R0 margins, and 64 Gy in 32 fractions of 2 Gy for patients with positive margins.

The delineation of target volumes (GTV, CTV, PTV) has already been described previously [15] (Figure 1). Irradiation of the entire circumference of the limb was avoided with the incorporation of a dosimetry-defined volume of normal tissue called the “skin corridor”. This corridor comprised at least 25% of the limb’s circumference, corresponding to approximately 10% of the cross-sectional area of the limb. The aim was to preserve lymphatic drainage and minimize the risk of lymphedema. The dose within the corridor was carefully restricted to no more than 35 Gy [17]. Dosimetric data of the skin corridor were available for 26 patients. The conformity index (CI) was calculated for all treatment plans with available data using the formula defined by ICRU 62 and originally utilized in the RTOG 90-05 study with a 95% isodose [18,19]: *CI* = *V_prescription_*/*V_target_*, where *V_prescription_* is the volume covered by the 95% prescription isodose surface, and *V_target_* is the target volume. A CI close to 1 is optimal. The homogeneity index (HI) was calculated using the following formula: *HI* = *(D_max_* − *D_min_)*/*D_prescription_*, where *D_max_* is the maximum dose received by any point within the target volume, *D_min_* is the minimum dose received by any point within the target volume, and *D_prescription_* is the prescribed dose for the target volume. An HI close to 0 is optimal.

### 2.3. Clinical Outcomes

The clinical outcomes comprised post-operative complications and radiation therapy (RT)-induced complications, graded in accordance with the National Cancer Institute Common Terminology Criteria for Adverse Events (CTCAE) version 5.0, and were defined as follows:(I)WCs, including infections, necrosis, and hematomas, occurring within 3 months post-surgery(II)Long-term lymphedema(III)Radiodermatitis and edema, which were considered acute complications related to RT(IV)Fibrosis, telangiectasias, and fractures, which were considered chronic complications related to RT(V)Amputations

### 2.4. Follow-Up

Outcome evaluation included weekly clinical examinations during radiotherapy, followed by assessments by a radiation oncologist, surgeon, and medical oncologist, if required, every 4–6 months for the first 5 years, and annually thereafter.

### 2.5. Statistical Analyses

Complication-free survival rates were calculated using the Kaplan–Meier method. Associations between variables were tested with the chi-squared test or Fisher’s exact test. An alpha risk of 5.0 was established. Prognostic factors showing a *p*-value < 0.1 in univariate analysis were included in a Cox regression model for multivariate analysis. Multicollinearity was assessed using the Belsley–Kuh–Welsch technique, and proportional hazards were evaluated with Schoenfeld residuals. A two-sided *p*-value < 0.05 was deemed statistically significant. Statistical analyses were conducted using EasyMedStat (version 3.30.2; Paris, France, www.easymedstat.com, accessed on 1 January 2024).

## 3. Results

### 3.1. This Patient Population

A total of 169 patients were enrolled between 2007 and 2020, with a median follow-up of 5.5 years for the entire cohort. Patient and tumor characteristics are detailed in Table 1. Most ESTSs were deep-seated (87.0%), situated in the lower arm (71.1%), and proximal (74.6%). Limb-sparing surgery was performed in 167 patients (98.8%), while two patients required amputation due to tumor extension. 

Treatment characteristic are detailed in Table 2. A total of 131 patients (77.5%) underwent surgical management at a referral center, and 38 patients (22.5%) underwent either biopsy excision or surgical management at a nonexpert center. Among all patients, the majority (91.7%) received RT postoperatively, and a smaller percentage (8.3%) received it preoperatively. The median duration from the onset of symptoms to the initial consultation was four months. For postoperative RT, the median time to start was 82 days, while for preoperative RT, the median interval between radiotherapy, and surgery was 83 days. Treatment characteristics, including median doses administered for both RT techniques, as well as the fractionation protocols, are detailed in Table 2. The median duration of radiotherapy delivery was 37 days. 

Thirty-five patients (20.7%) received treatment with 3DRT, one was treated with 2DRT, and 134 patients (79.3%) received IMRT. The median D_98%_ and V_95%_ were significantly greater for IMRT than for 3DRT (96.6% vs. 87.1%; *p* < 0.001, and 99.2% vs. 92.0%; *p* < 0.001, respectively). Conversely, the median D_2%_ was lower for IMRT (103.2% vs. 105.6%; *p* < 0.001). The median CIs were 1.24 for IMRT and 1.62 for 3DRT (*p* = 0.038), while the median HIs were 0.07 for IMRT and 0.26 for 3DRT (*p* < 0.001).

Long-term local control, distant control, and overall survival were described in a previous article [15].

### 3.2. Postoperative Complications

WCs occurred in 38 patients (22.5%). Among them, 30 patients (84.2%) developed one WC, and 8 (15.8%) developed at least two WCs. The WC rates were 20.7% and 42.9% for patients treated with postoperative and preoperative RT, respectively (OR = 2.88; CI [0.93; 8.9]; *p* = 0.088). The results of complications are presented in Table 3. The most frequent complication was postoperative infection requiring antibiotic therapy, which was observed in 25 (14.9%) patients. Hematomas were present in 17 (10.1%) patients, and necrosis was observed in 8 patients (4.8%). Vacuum-assisted closure (VAC) dressing was utilized for 52 patients (31.0%). Among them, VAC was used postoperatively in 20 patients (38.5%) due to WCs, while VAC was utilized intraoperatively in 32 patients (61.5%) because closure of the surgical planes was not feasible. The VAC rates were 47.4% and 26.2% in patients who did and did not develop WCs, respectively (OR = 2.54; CI [1.2; 5.37]; *p* = 0.022). Skin grafts were used for 33 patients (19.5%). Excluding patients requiring reoperation for positive margins, 56 patients (33.1%) underwent reoperation, which was directly due to a postoperative complication for 26 patients (15.4%) or due to impossible primary skin closure necessitating subsequent revision of VAC application and skin grafting.

According to the univariate analysis, patients with ESTSs located in the upper limbs had significantly fewer WCs than those with tumors located in the lower limbs (7.9% vs. 92.1%; OR = 0.14; CI [0.042; 0.49]; *p* = 0.001). A tumor size ≤ 5 cm was significantly associated with a lower WC rate (13.2% vs. 86.8%; OR = 0.29; CI [0.11; 0.79]; *p* = 0.02). The median tumor size in patients who presented with a WC was 13.7 cm (range 0.4–30) versus 9.0 cm (2–60) for those who did not (*p* = 0.016). The WC rate was lower for Grade 2–3 tumors than for Grade 1 tumors (47.4% vs. 52.6%; OR = 0.23; CI [0.11; 0.51]; *p* < 0.001). Higher WC rates were found for proximal ESTSs than for distal ESTSs (94.7% vs. 5.3%; OR = 8.2; CI [1.88; 35.7]; *p* = 0.002) and for liposarcomas than for other pathologies (60.6% vs. 39.5%; OR = 4.55; CI [2.13; 9.74]; *p* < 0.001).

WC rates did not significantly differ according to age, sex, depth, margin status, second surgery for margin status, RT schedule, RT dose, skin corridor dose, or the use of neoadjuvant or concomitant chemotherapies. Thirty-eight patients (22.5%) were not directly referred to a reference center. These patients underwent surgery either definitively at a nonspecialized center or initially at a nonspecialized center and subsequently at a reference center for surgical revision. No significant relationship was found between the WC rate and surgery in a specialized center (OR = 2.22; CI [0.8; 6.16]; *p* = 0.129).

According to the multivariate analysis (Table 4), upper limb localization (OR = 0.2, CI [0.057; 0.705], *p* = 0.012) and distality (OR = 0.188 [0.041; 0.851]; *p* = 0.03) were favorable prognostic factors for WCs. VAC utilization was identified as an unfavorable risk factor for WCs (OR = 3.65, CI [1.56; 8.56], *p* = 0.003), although 38.5% of VAC procedures were performed following WCs. Size, grade, and liposarcoma status were not significant risk factors for WCs. No significant difference in OS was found between patients who experienced WCs and those who did not (*p* = 0.139).

Lymphedema was diagnosed in 12 patients (7.1%), without significant preferential localization. No significant associations were found between lymphedema and age, sex, size, depth, WC rate, VAC use, RT dose, RT schedule, PTV, acute or chronic RT-induced side effects, or CT use. Only one patient with lymphedema had dosimetric data for the skin corridor.

### 3.3. Acute Complications Related to Radiotherapy

Radiodermatitis was the most prevalent adverse event, which occurred in 131 patients (78%), 43.1% of whom had grade ≥ 2 radiodermatitis, while 7.1% had a grade ≥ 3 radiodermatitis. One patient had grade 4 radiodermatitis. Higher rates of radiodermatitis were associated with superficial tumors, with all 23 patients with superficial tumors exhibiting radiodermatitis (*p* = 0.003). Similarly, the IMRT technique was associated with a higher incidence of radiodermatitis than 3DRT, with rates of 85.5% and 13.7%, respectively, within each group for patients with radiodermatitis (*p* = 0.001).

In contrast, sex, age, site, localization, tumor size, histology, grade, margin status, WC rate, VAC use, surgery in a reference center, second surgery, CT use, and PTV size were not associated with radiodermatitis. RT dose and RT schedule (normo-, hypo-, and hyper-fractionation) did not specifically affect the radiodermatitis rates. The median BED to the PTV was similar between patients who did and did not have radiodermatitis (82.8 BEDGy_4_ vs. 85.1 BEDGy_4_ and 66.1 BEDGy_10_ vs. 67.9 BEDGy_10_).

Acute edema was diagnosed in 57 patients (34.1%). Only 3 (1.8%) patients had grade ≥ 2 edema. Edema developed significantly more frequently with VAC use (42.1% vs. 24.7%; OR = 2.21; CI [1.12; 4.37]; *p* = 0.034). Patients who experienced edema had higher rates of second surgery than those without edema (47.3% vs. 28.2%; OR = 2.29; CI [1.18; 4.46]; *p* = 0.022). Edema was associated with a greater percentage of hematomas (17.54% vs. 6.42%; OR = 3.1; CI [1.11; 8.65]; *p* = 0.032). Sex, age, localization, tumor size, depth, histology, grade, margin status, surgery in a reference center, skin grafting, RT dose, RT, schedule, PTV, skin corridor dose, and RT technique were not linked to the occurrence of edema. No significant risk factors were identified in the multivariate analysis.

### 3.4. Chronic Complications Related to Radiotherapy

Telangiectasias occurred in 21.7% of patients. PTV doses ≥ 85 BEDGy_4_ and ≥62 BEDGy_10_ were significantly associated with increased telangiectasis rates, with 63.9% of patients exhibiting telangiectasis and 36.9% not exhibiting telangiectasis (OR = 3.02; CI [1.4; 6.51]; *p* = 0.007). The median doses were 88.2 BEDGy_4_ and 70.1 BEDGy_10_ for patients with telangiectasis and 82.1 BEDGy_4_ and 65.5 BEDGy_10_ for patients without telangiectasis (*p* = 0.014). Therefore, incomplete margin status was also more strongly associated with telangiectasias than R0 status (61.1% vs. 38.9%; *p* < 0.001). Grade 1 tumors were more strongly associated with telangiectasias than were grade 2 and 3 tumors (OR = 0.28; CI [0.12; 0.63]; *p* = 0.003). No relationship was found between skin corridor dose and telangiectasia. No risk factor for telangiectasias retained significance on multivariate analysis.

Fibrosis occurred in 38.7% of patients, with only one case of grade 3 fibrosis. A tumor size ≥ 10 cm was associated with higher rates of fibrosis, with 50.8% and 34.0% of patients, respectively, having fibrosis (OR = 2.0; CI [1.06; 3.78]; *p* = 0.046). Patients who experienced fibrosis had higher rates of liposarcoma than patients with other pathologies (43.2% vs. 23.3%; OR = 3.19; CI [1.64; 6.22]; *p* < 0.001) and higher rates of low-grade tumors (42.9% vs. 19.0%; OR = 0.31; CI [0.1; 0.64]; *p* = 0.002). Patients with complete margins after surgery had lower fibrosis rates than those with incomplete margins (52.3% vs. 72.0%; OR = 0.43; CI [0.22; 0.82]; *p* = 0.016). Patients who developed acute edema had significantly higher rates of fibrosis (46.2 vs. 26.5; OR = 2.38; CI [1.23; 4.59]; *p* = 0.014).

Patients with fibrosis were significantly more likely to be treated with IMRT than with 3DRT (92.3% vs. 7.7%), while among those who did not develop fibrosis, the percentages were 71.8% and 27.2%, respectively (OR = 0.22; CI [0.08; 0.61]; *p* = 0.003). The median PTV was significantly greater in patients with fibrosis (1096.2 cc, range 87.1–4901.9) than in those without fibrosis (584.9 cc, range 57.7–8714.4), with a median difference of 511.3 cc (*p* = 0.001). Additionally, patients with PTVs less than 750 cc exhibited a lower fibrosis rate (37.5% vs. 62.5%; OR = 0.37; CI [0.19; 0.7]; *p* = 0.004). When the PTVs were analyzed in quartiles, the proportions of patients in the Q1 (57.7–370.4 cc), Q2 (370.5–699.0 cc), Q3 (699.1–1504.6 cc), and Q4 (1504.7–8714.4 cc) groups who had fibrosis were 18.7%, 18.7%, 25.0%, and 37.5%, respectively (*p* = 0.016).

CT administered preoperatively, concomitantly, or postoperatively was not associated with the development of fibrosis (*p* = 0.082). 

A trend toward higher mean doses to the skin corridor was observed in patients who developed fibrosis: 19.95 Gy (EQD2) (SD 7.51) compared to 15.45 Gy (EQD2) (SD 2.88); CI = [−0.075; 9.07]; *p* = 0.062. No differences were observed between these groups in terms of V_10Gy_, V_15Gy_, V_20Gy_, V_25Gy_, or V_30Gy_. 

Other tumor, patient, and treatment characteristics including RT schedule and RT fractionation were not associated with the occurrence of telangiectasis or fibrosis. 

Fractures were observed in 5 patients (3%), including a femur fracture in two patients, a tibial fracture in one patient, a pelvis fracture in one patient, and a metacarpal fracture in one patient, with a median time of onset of 4.9 years (range 0.9–9.7). At 5 years, the fracture-free survival rate was 97.4% (95% CI: 92.0–99.2). All fractures occurred in patients with deep tumors. The distribution of fractures was not significantly different between the upper limbs and the lower limb (1 vs. 4 patients; *p* > 0.999) or between proximal and distal locations (3 vs. 2 patients; *p* = 0.603). Although two patients were older than 60 years and three were younger than 60 years, age did not exhibit a significant association with fractures (*p* = 0.402). The sex distribution (four males and one female) also did not reach statistical significance (*p* = 0.202). The median size of the tumors did not differ significantly between nonfracture patients and fracture patients (9.9 cm vs. 12.5 cm; *p* = 0.614). WCs were observed in two of five fracture cases (*p* = 0.278).

The median doses to the PTV for nonfracture patients were 75.0 BEDGy_4_ and 60.0 BEDGy_10_, while fracture patients received median doses of 96.0 BEDGy_4_ and 76.8 BEDGy_10_, although these differences did not reach statistical significance (*p* = 0.311). All fractures occurred in patients who had received normofractionated IMRT. PTVs were not significantly different, with 1522 cc for fracture patients and 1054 cc for nonfracture patients (*p* = 0.382). The D_2%_ and D_98%_ values were similar between the two groups (*p* = 0.761). No significant difference in the V_40Gy_ to the bone was identified, with rates of 29% for nonfracture patients and 37% for fracture patients (*p* = 0.414). A V_40Gy_ < 64% in the main irradiated bones was used for every fracture patient. However, when comparing IMRT and RT3D, the V_40Gy_ was significantly greater in the RT3D group than in the IMRT group (41.8% vs. 28.1%; *p* = 0.043). Additionally, adherence to the V_40Gy_ dose constraint < 64% was significantly lower in the RT3D group than in the RCMI group, with rates of 9.6% and 90.4%, respectively (*p* = 0.014). The two patients with femoral fractures received a dose exceeding 59 Gy (88.5 BEDGy_4_ and 70.8 BEDGy_10_). CT use was not significantly different between the two groups (*p* > 0.999).

Amputation of the lower limbs was performed for 6 patients. Four amputations were performed as a salvage treatment for local relapse, and two were performed initially before RT due to the large sizes of the tumors; these patients were not eligible for limb-sparing surgery. No complications resulted in requiring amputation in the current series.

## 4. Discussion

The authors of this study significantly contribute to the understanding of long-term complications associated with multimodal treatments involving both surgery and pre- or post-operative RT for ESTSs. One of the key strengths of this study is its meticulous examination of postoperative complications, acute and long-term complications related to RT, and fractures, offering a holistic view of treatment-related adverse events. This comprehensive analysis sheds light on various factors influencing treatment outcomes and provides valuable insights for clinical practice.

O’Sullivan et al. previously demonstrated in a robust study that preoperative RT was significantly associated with more postoperative WCs, while postoperative RT was more strongly associated with long-term complications such as fibrosis or joint stiffness [6]. In a recent systematic review of studies on ESTSs [12], the WC rate ranged from 8% to 41% for patients treated with preoperative RT and from 2% to 27% for patients treated with postoperative RT, with an overall median rate of 18%. In this series, WCs occurred in 22.5% of patients, which is consistent with the literature. This higher incidence of WCs associated with preoperative RT reported in the literature may have dissuaded surgeons from proposing this treatment, which may explain why only 14 patients were treated with this schedule in the current series. However, despite this low number, we observed a trend toward higher WC rates for patients who received preoperative RT versus those who received postoperative RT, with 42.9% vs. 20.7%, respectively (OR = 2.88; 95% CI [0.93; 8.9]; *p* = 0.088). The lack of statistical significance could be due to insufficient power secondary to the low number of patients. When excluding reoperation for VAC revision and skin grafting due to impossible primary skin closure, 15.4% of patients required secondary surgical intervention following WCs, which is similar to the findings of a comparable study published by Ouyang et al., where 19% of patients required secondary surgical intervention following WCs [20]. However, the median time between preoperative radiotherapy and surgery was 83 days, a delay longer than the 3 to 6 weeks reported in the literature [6,21,22]. This delay may be due to complications, as well as delays in the scheduling of treatments and presentations at multidisciplinary meetings.

In this series, ESTSs located proximally and in the lower extremities were identified as a risk factor for developing WCs. These findings align with those reported by Talbert et al. who reported higher complication rates among patients with lesions in the lower extremities than among those with lesions in the upper extremities [23], and by Cannon et al. who noted a higher incidence of chronic WCs for patients with proximal tumors than for those with distal tumors [14]. These results may be attributed to the larger sizes of tumors located in the lower extremities, which necessitate more extensive surgical interventions involving larger operating beds, thus increasing the likelihood of experiencing complications [24]. Similarly, in the present study, a tumor size exceeding 5 cm was associated with a higher WC rate according to univariate analysis, which is consistent with the findings reported by Cannon et al. (*p* = 0.035) [14]. Furthermore, other authors have reported a significant association by multivariate analysis for tumors larger than 10 cm (OR = 1.55, 95% CI 0.78–3.11), as reported in a meta-analysis by Slump et al. [25].

Compared with superficial tumors, deep-seated tumors have been found to be associated with higher WC rates only in univariate analyses conducted in studies involving 728 and 191 patients [26,27]. However, these associations were not observed in our study. This discrepancy may be attributed to limited statistical power, as only 13.6% of the tumors in our study were superficial. In contrast, Baldini et al. reported a significant association between WCs and superficial tumors in patients treated with preoperative RT [28]. They explained that tumors located close to the skin surface result in the overlying skin and nearby subcutaneous tissues, which are crucial for wound closure, receiving the full dose of RT. Another possibility may be the higher incidence of postoperative infection for superficial tumors, mainly due to the presence of *Staphylococcus aureus*, a bacterial species present on the skin surface [29]. However, their results pertained to patients who were all treated with preoperative RT unlike in the current study.

Cheng et al. demonstrated a significant association between WCs and lower overall survival [30], which was not observed in this study. The authors did not analyze the localization or size of tumors in patients who experienced WCs. Conceivably, large proximal tumors may lead to more severe complications, subsequently resulting in greater limitations in daily life and potentially impacting survival. However, in this series, WCs appeared to be transient and well managed with pharmacological or surgical intervention.

As various authors have already demonstrated, the management of ESTSs, including complications and oncological outcomes, is better in specialized sarcoma centers [5,31,32,33,34]. WCs appear to be more frequent in nonspecialized centers, likely due to the experience of the surgeons. However, we could not demonstrate this difference. Our results are likely biased due to the lack of data collected from postoperative patients in nonexpert centers.

Regarding RT-related adverse events, significant associations were found between superficial tumors and radiodermatitis, between VAC use or second surgery and edema, between telangiectasias and both incomplete margins and dose, and between fibrosis and tumor size ≥ 10 cm, PTV ≥ 750 cc, incomplete margins, liposarcomas, and grade 1 pathology in the univariate analysis. According to the multivariate analysis, only PTVs ≥ 750 cc were associated with risk factors for fibrosis (*p* = 0.0316). Higher doses administered to patients with positive margins were not found to be a significant risk factor for telangiectasias, although a trend was observed (*p* = 0.0565). The relationship between the highest rates of telangiectasias and grade 1 sarcomas rather than grade 2 and 3 sarcomas may be explained by a competing risk because telangiectasias appear later, and Grade 2 and 3 cases are more likely to recur and lead to death more quickly. Folkert et al. demonstrated a decrease in radiodermatitis and edema rates with IMRT compared to 3DRT in a series of 319 patients [13]. The authors found rates of 39.8% and 11.3% for grade ≥ 2 radiodermatitis and edema, respectively. The results of the present study are consistent with these rates, with 43.2% of patients having grade ≥ 2 radiodermatitis and 1.8% having grade ≥ 2 edema. In the present study, if radiotherapy-related complications were more common for patients treated with RCMI, the grading of side effects using CTCAE was frequently underutilized for the initial patients in the series treated with 3DRT.

A better V_40Gy_ < 64% to the bone was achieved in patients treated with IMRT, a threshold previously shown to be a risk factor for long bone fractures [35,36]. Only 14 patients (9.3%) had a V_40Gy_ > 64% in the population, resulting in a total of 5 fractures (3%), which is comparable to the overall fracture rate of 5.3% reported in the literature [12]. The median time to fracture in our previous systemic literature review ranged from 3.2 years to 7.3 years [12,13,14,37], which is similar to the median time of 4.9 years (range 0.9–9.7) reported in the current study. Stewart et al. also showed that IMRT appears to be a good option for treatment due to its better sparing of healthy tissue and lower femur V_45Gy_ [38]. The median CI was lower in the IMRT group than in the 3DRT group (1.24 vs. 1.62), indicating an increase in the target dose distribution with a concurrent decrease in high doses to normal tissue. Consequently, IMRT appears to be the treatment of choice for this pathology [39].

The limitations of the current study include its retrospective nature and the limited documentation of adverse events during the era of 3DRT, which was either due to incomplete data collection or inadequate grading according to the CTCAE classification. Additionally, no significant risk factor was identified for fractures.

Furthermore, the small number of patients treated with preoperative RT and 3DRT did not permit comparisons of groups in equal proportions, thus limiting the ability to draw definitive conclusions for such comparisons, although relevant values were identified for IMRT, allowing future comparisons in prospective trials.

## 5. Conclusions

This study provides a comprehensive examination of both short- and long-term postoperative and post-RT complications in patients treated with ESTS. The excellent overall tolerance of multimodal treatment combining surgery and RT was demonstrated, with acceptable and manageable side effects, highlighting the capacity of modern treatment modalities to achieve optimal outcomes while minimizing treatment-related morbidity. Numerous risk factors have been highlighted, providing valuable insights for optimizing treatment strategies and enhancing patient care for this rare disease.

## Figures and Tables

**Figure 1 cancers-16-01977-f001:**
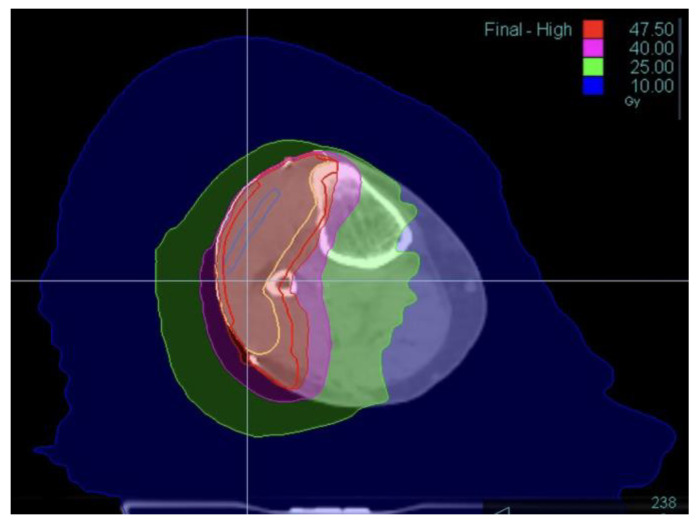
Dosimetry of postoperative radiotherapy for a soft tissue sarcoma (STS) located on the outer side of the knee.

**Table 1 cancers-16-01977-t001:** Patient and tumor characteristics.

	Total Patients = *n* (%)	Postoperative RT = 155 (91.7%)	Preoperative RT = 14 (8.3%)	*p* Value
**Sex**				**0.048**
Male	83 (49.1)	80 (51.6)	3 (21.4)	
Female	86 (50.9)	75 (48.4)	11 (78.6)	
**Age** (median)	64 (21–94)	65 (21–94)	57 (37–72)	0.264
>60 years	100 (59.2)	94 (60.6)	6 (42.9)	
<60 years	69 (40.8)	61 (39.3)	8 (57.1)	
**Location**				
Upper arm	52 (30.8)	48 (31.0)	10 (71.4)	>0.999
Lower arm	117 (69.2)	107 (69.0)	4 (28.6)	
Proximal	126 (74.6)	114 (73.5)	12 (85.7)	0.522
Distal	43 (25.4)	41 (26.5)	2 (14.3)	
**Depth**				0.221
Superficial	23 (13.6)	23 (14.8)	0 (0.0)	
Deep	146 (86.4)	132 (85.2)	14 (100)	
**Grade (FNCLCC)**				0.1
1	45 (26.6)	38 (24.5)	7 (50.0)	
2	44 (26.0)	40 (25.8)	4 (28.6)	
3	70 (41.4)	67 (43.2)	3 (21.4)	
Unknown	10 (5.9)	10 (6.5)	0 (0.0)	
**Margins status**				0.202
R0	106 (62.7)	98 (63.5)	8 (57.4)	
R1	57 (33.7)	54 (34.9)	3 (21.4)	
R2	3 (1.8)	2 (1.3)	1 (7.1)	
Unknown	3 (1.8)	2 (1.3)	1 (7.1)	
**Size**				
Median size (cm)	8.0 (0.4–60)	8 (0.4–39)	9.3 (3–60)	0.28
<10	101 (59.8)	94 (60.7)	7 (50.0)	
≥10	68 (40.2)	61 (39.4)	7 (50.0)	
**TNM**				0.843
T1a	15 (8.9)	15 (9.7)	0 (0.0)	
T1b	25 (14.8)	23 (14.8)	2 (14.3)	
T2a	12 (7.1)	11 (7.1)	1 (7.1)	
T2b	117 (69.2)	106 (68.4)	11 (78.6)	
**Histological subtype**				0.404
Liposarcoma	56 (33.1)	48 (31.0)	8 (57.1)	
UPS	31 (18.3)	30 (19.4)	1 (7.1)	
Myxofibrosarcoma	32 (18.9)	28 (18.1)	4 (28.6)	
Leiomyosarcoma	17 (10.0)	17 (11.0)	0 (0)	
Synovial sarcoma	6 (3.6)	6 (3.9)	0 (0)	
Undifferentiated sarcoma	9 (5.3)	8 (5.2)	1 (7.1)	
Rhabdomyosarcoma	4 (2.4)	4 (2.6)	0 (0)	
Other	14 (8.3)	14 (9.0)	0 (0)	

Values in bold mean significance (*p* < 0.05). UPS = undifferentiated pleomorphic sarcoma.

**Table 2 cancers-16-01977-t002:** Treatment characteristics.

	Total Patients = *n* (%)	Postoperative RT = 155 (91.7%)	Preoperative RT = 14 (8.3%)	*p* Value
**RT Technique**				0.152
2DRT	1 (0.6)	1 (0.7)	0 (0)	
3DRT	34 (20.1)	34 (21.9)	0 (0)	
IMRT	134 (79.3)	120 (77.4)	14 (100)	
**BEDGy_4_**				
BEDGy_4_ (median)	75.0 (45.0–109.9)	75.0 (45.0–109.9)	75.0 (73.1–99.8)	**0.04**
BED_4_ ≥ 75 Gy	163 (96.4)	150 (96.8)	13 (92.9)	0.41
BED_4_ < 75 Gy	6 (3.6)	5 (3.2)	1 (7.1)	
BED_4_ ≥ 85 Gy	72 (42.6)	70 (45.2)	2 (14.3)	**0.027**
BED_4_ < 85 Gy	97 (57.4)	85 (54.8)	12 (85.7)	
BED_4_ ≥ 95 Gy	52 (30.8)	50 (32.3)	2 (14.3)	0.231
BED_4_ < 95 Gy	117 (69.2)	105 (67.7)	12 (85.7)	
**BEDGy_10_**				
BEDGy_10_ (median)	60.0 (21.45–87.20)	60 (36.00–87.20)	60 (57.4–76.8)	**0.026**
BED_10_ ≥ 60 Gy	159 (94.0)	148 (95.5)	13 (92.9)	0.507
BED_10_ < 60 Gy	10 (6.0)	7 (4.5)	1 (7.1)	
BED_10_ ≥ 65 Gy	72 (42.6)	70 (45.2)	2 (14.3)	**0.027**
BED_10_ < 65 Gy	97 (57.4)	85 (54.8)	12 (85.7)	
**Fractionation (Gy per day)**				>0.999
Normo (1.8–2 Gy)	154 (91.2)	141 (91.0)	13 (92.9)	
Hypo (3 Gy)	13 (8.7)	12 (7.4)	1 (7.1)	
Hyper (1.15 Gy × 2)	2 (1.2)	1 (1.3)	0 (0.0)	
**Tumor volumes**				
GTV (cm^3^)	213.4	206.0	250.8	0.176
CTV (cm^3^)	446.6	417.7	1014.4	**0.039**
PTV (cm^3^)	699.0	672.8	1291.6	0.078
**Chemotherapy**				
CT (+)	43 (25.4)	35 (22.6)	8 (57.1)	**0.009**
CT (−)	126 (74.6)	120 (77.4)	6 (42.9)	
Neoadjuvant	16 (37.2)	9 (25.7)	7 (87.5)	**0.003**
Adjuvant pre-RT	19 (44.2)	19 (54.3)	0 (0)	**0.022**
Concomitant	4 (9.3)	4 (11.4)	0 (0)	>0.999
Adjuvant post-RT	4 (9.3)	3 (8.6)	1 (12.5)	0.556
**Coverage**				
	**All patients**	**IMRT**	**3DRT**	
**PTV D_98%_** (median)	96.3 (59.6–98.6)	96.6 (59.6–98.6)	87.1 (63.3–97.2)	**<0.001**
**PTV D_2%_** (median)	103.4 (100.3–108.9)	103.2 (100.3–107.0)	105.6 (103.3–108.9)	**<0.001**
**PTV V_95%_** (median)	99.1 (61.0–99.99)	99.2 (75.8–99.99)	92 (61.0–98.8)	**<0.001**
**CI** (median)	1.25 (1.01–2.38)	1.24 (1.01–2.38)	1.62 (1.05–2.23)	**0.038**
**HI** (median)	0.08 (0.03–0.46)	0.07 (0.03–0.46)	0.26 (0.07–0.42)	**<0.001**

Values in bold mean significance (*p* < 0.05). BED = biological effective dose; CI = conformity index; CT = chemotherapy; CTV = clinical target volume; GTV = growth tumor volume; HI = homogeneity index; IMRT = intensity-modulated radiation therapy; PTV = planning target volume; RT = radiotherapy; 2DRT = two-dimensional conformal radiotherapy; 3DRT = three-dimensional conformal radiotherapy.

**Table 3 cancers-16-01977-t003:** Complications.

	All Patients		Postoperative RT		Preoperative RT		*p* Value
	Patients (*n*)	%	Patients (*n*)	%	Patients (*n*)	%	
	169	100	155	91.7	14	8.3	
**Wound complications**	38	22.5	32	20.7	6	42.9	0.088
Hematoma	17	10.1	13	8.4	4	28.6	**0.038**
Necrosis	8	4.8	7	4.6	1	7.1	0.509
Infection	25	14.9	22	14.3	3	21.4	0.441
**Lymphoedema**	12	7.1	9	5.5	3	21.4	0.065
**Acute radiation-related complications**	136	81.0	136	82.9	9	64.3	0.146
**Radiodermatitis**	131	78.0	132	80.5	8	57.1	0.084
G0	37	22.0	32	19.5	6	42.9	
G1	68	40.5	68	41.5	4	28.6	
G2	50	29.8	50	30.5	2	14.3	
G3	12	7.1	13	7.9	2	14.3	
G4	1	0.6	1	0.6	0	0.0	
**Edema**	57	34.1	54	33.1	5	35.7	>0.999
G0	110	65.9	109	66.9	9	64.3	
G1	54	32.3	50	30.7	5	35.7	
G2	3	1.8	3	1.9	0	0.0	
**Chronic radiation-related complications**	75	44.6	71	43.3	7	50.0	0.065
**Telangiectasias**	36	21.7	34	21.0	2	14.3	0.737
G0	130	78.3	128	79.0	12	85.7	
G1	22	13.3	20	12.4	2	14.3	
G2	13	7.8	13	8.0	0	0.0	
G3	1	0.6	1	0.6	0	0.0	
**Fibrosis**	65	38.7	62	37.8	6	42.9	0.779
G0	107	61.3	102	62.2	0	57.1	
G1	38	22.6	34	20.7	5	35.7	
G2	26	15.5	27	16.5	1	7.1	
G3	1	0.6	1	0.6	0	0.0	
**Fracture**	5	3.0	5	3.1	0	0.0	>0.999
Femur	2	40.0	2	40.0	0	0.0	
Tibia	1	20.0	1	20.0	0	0.0	
Pelvis	1	20.0	1	20.0	0	0.0	
Metacarpus	1	20.0	1	20.0	0	0.0	
**Amputation**	6	3.6	4	2.6	2	14.3	0.08

Values in bold mean significance (*p* < 0.05). G = grade.

**Table 4 cancers-16-01977-t004:** Multivariate analysis of WCs risk factors.

Wound Complications	Odds Ratio	*p* Value
** Site ** (Upper limbs)	0.201 [0.0557; 0.728]	** 0.0145 **
** Localization ** (Distal)	0.127 [0.0264; 0.613]	** 0.0102 **
** VAC **	3.65 [1.56; 8.56]	** 0.00285 **
** Size ** ** ≤ ** ** 5 cm **	0.53 [0.174; 1.61]	0.262
** Liposarcoma **	2.72 [0.99; 7.49]	0.0524
**Grade** (I)	2.17 [0.779; 6.04]	0.138
**Edema**	** Odds Ratio **	** * p * ** ** Value **
** VAC **	1.64 [0.428; 6.28]	0.471
** Second surgery post WC **	1.27 [0.331; 4.9]	0.725
** Hematoma **	2.52 [0.838; 7.56]	0.1
**Telangiectasias**	** Odds Ratio **	** * p * ** ** Value **
**Grade** (I)	2.07 [0.82; 5.21]	0.124
** Negative margins **	0.374 [0.136; 1.03]	0.0565
** BEDGy_4_ ** ** ≥ ** ** 85 Gy **	1.8 [0.613; 5.27]	0.286
**Fibrosis**	** Odds Ratio **	** * p * ** ** Value **
** Size ** ** ≥ ** ** 10 cm **	1.08 [0.468; 2.5]	0.854
** Liposarcoma **	1.26 [0.437; 3.64]	0.668
**Grade** (I)	2.06 [0.677; 6.28]	0.203
** Negative margins **	0.522 [0.26; 1.05]	0.0681
** PTV < 750 cc **	0.402 [0.177; 0.914]	** 0.0298 **

Values in bold mean significance (*p* < 0.05). BED = biological effective dose; PTV = planning target value; VAC = vacuum-assisted closure; WCs = wound complications.

## Data Availability

The data are available upon request due to privacy restrictions. The data presented in this study are available upon request from the corresponding author.

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
