# Peer review of "Complications and Risk Factors in Patients with Soft Tissue Sarcoma of the Extremities Treated with Radiotherapy"

_cancers, 2024, doi:10.3390/cancers16111977_

Round 1

Reviewer 1 Report

Comments and Suggestions for Authors

This manuscript explores the various challenges associated with treating soft tissue sarcoma using multimodal radiotherapy. The study primarily aims to evaluate the toxicities associated with multimodal treatment, including neoadjuvant radiochemotherapy and adjuvant radiotherapy. The authors have done an excellent job of describing their methods and presenting their results, making the discussion very informative.

I have the following note to the authors to be improved:
1) add more details about how the different radiation techniques and doses specifically affect the outcomes, which could help in tailoring patient-specific treatment plans. Also the authors didn't described their radiotherapy protocolls in the material and methodes, single dose and the whole doses.

2) they should discuss the long-term effects of multimodal treatment on patients, including their quality of life and survival rates. This could add another layer of depth to the study and make it more comprehensive.

Author Response

Dear Reviewer,
Thank you for your insightful comments on our study on complications of soft tissue sarcoma of the extremities treated with radiotherapy.

1) add more details about how the different radiation techniques and doses specifically affect the outcomes, which could help in tailoring patient-specific treatment plans. Also the authors didn't described their radiotherapy protocolls in the material and methodes, single dose and the whole doses.

ResponseRegarding radiotherapy regimens, there was a multitude of protocols used, especially during the era of 3D radiotherapy. Since the introduction of IMRT, the regimens have become more standardized, with the main ones being 50 Gy in 25 fractions of 2 Gy and 64 Gy in 33 fractions of 2 Gy. Therefore, we converted all doses to BED to facilitate dose comparisons between the regimens.
Different radiation techniques (fractionation, timing, IMRT/3D-RT), and doses were analyzed for each complication (post-surgery and post-radiotherapy in both the short and long term) using univariate and multivariate analyses where possible. Significant and non-significant associations are described for each part.
All these modifications described here have been incorporated into the text.

2) they should discuss the long-term effects of multimodal treatment on patients, including their quality of life and survival rates. This could add another layer of depth to the study and make it more comprehensive.

Response : Long term local control, distant control and overall survival were described in another article : https://doi.org/10.3390/cancers16101789. The reference had been added in the text.

Reviewer 2 Report

Comments and Suggestions for Authors

This retrospective study aimed to assess complications associated with multimodal treatments involving surgery and radiotherapy (RT) and to identify potential risk factors.

The topic is interesting and the paper well organized.

Abstract: "proximal" location should be detailed.

Irradiation technique and field: please provide an image explaining the fields.

Please discuss indication to RTE in R0 and R2 margins. 

Was local control correlated to the rate of major complications?

Any fracture?Any prophylactic osteosynthesis?Please discuss

Author Response

Dear Reviewer,
Thank you for your thoughtful review of our study on complications of extremity soft tissue sarcoma.
1) Abstract: "proximal" location should be detailed.
Response : Proximal location has been detailed in the abstract " ESTSs primarily occurred proximally (74.6%) which is defined as above the knees and elbows."

2) Irradiation technique and field: please provide an image explaining the fields.
Response : An image had been added as an exemple: 
Figure 1. Dosimetry of postoperative radiotherapy for a soft tissue sarcoma (STS) located on the outer side of the knee.

3) Please discuss indication to RTE in R0 and R2 margins. 
ResponseRT for R0 was prescribed after multidisciplinary discussion considering is the following factors were present: tumor size exceeding 5 cm, grade 2-3, deep tumor. Curative RT was prescribed for patients with R2 margins following maximal resection by the surgeon, and further excision is not feasible.

4) Was local control correlated to the rate of major complications?
ResponseLong term local control, distant control and overall survival were described in another article : https://doi.org/10.3390/cancers16101789. Reference had been added in the text.
No, at 5 years, the LC rates was 92.9% (95% CI: 86.3-96.4) for patient without postoperative complication and 88.2% (95% CI: 71.2-95.5) for patients with postoperative complications. (p = 0.262.)

5) Any fracture?Any prophylactic osteosynthesis?Please discuss
Response : Fractures and risk factor were discussed in the results : "Fractures were observed in 5 patients (3%), including a femur fracture in two patients, a tibial fracture in one patient, a pelvis fracture in one patient, and a metacarpal fracture in one patient, with a median time of onset of 4.9 years (range 0.9–9.7). "
We did not identify any significative dose-specific risk factor for fracture, so prevention was not possible for these patients. Moreover, there was no specific bone involvement that could "predict" a risk of fracture.

Reviewer 3 Report

Comments and Suggestions for Authors

Great manuscript. Only one comment:

There seems to be conflicting findings reported on radio-dermatitis for IMRT vs 3d. On line 244, under section "3.3. Acute complications related to radiotherapy" the authors reported IMRT resulting in HIGHER dermatitis incidents than 3D but in line 407 of the discussion session, the authors reported, backed by current literature with LOWER radio-dermatitis rate for IMRT vs 3D. Also, please add a tab before 'Folkert et al.'

Author Response

Dear reviewer, 
Thank you for your thoughtful review of our study on complications of soft tissue sarcoma of the extremities.
We actually observed higher rates of radiodermatitis for patients treated with IMRT than with RT3D.  We tried to give an explanation in the discussion:
"In the present study, if radiotherapy-related complications were more common for patients treated with RCMI, the grading of side effects using CTCAE was frequently underutilized for the initial patients in the series treated with 3DRT." These results were in contrast with those reported by Folkert et al. who found lower rates of radiodermatitis with IMRT vs RT3D.
A tab has been added before Folkert et al.

Round 2

Reviewer 1 Report

Comments and Suggestions for Authors

The authors have provided the additional data and made the necessary corrections to their manuscript based on the reviewers' notes. Therefore, I suggest accepting this manuscript.